# HUMAN BASELINES IN MODEL EVALUATIONS NEED RIGOR AND TRANSPARENCY

**Kevin L. Wei**[*]
Harvard University

**Patricia Paskov**[*]
Independent

**Sunishchal Dev**[*]
Algoverse

**Michael J. Byun**[*]
Independent

**Anka Reuel**
Stanford University

**Xavier Roberts-Gaal**
Harvard University

**Rachel Calcott**
Harvard University

**Evie Coxon**
Max Planck School of Cognition

**Chinmay Deshpande**
Center for Democracy and Technology

## ABSTRACT

**This position paper argues that human baselines in foundation model evaluations must be more rigorous and more transparent to enable meaningful comparisons of human vs. AI performance.** Human performance baselines are vital for the machine learning community, downstream users, and policymakers to interpret AI evaluations. Models are often claimed to achieve "super-human" performance, but existing baselining methods are neither sufficiently rigorous nor sufficiently well-documented to robustly measure and assess performance differences. Based on a meta-review of the measurement theory and AI evaluation literatures, we derive a framework for assessing human baselining methods. We then use our framework to systematically review 113 human baselines in foundation model evaluations, identifying shortcomings in existing baselining methods. We publish our framework as a reporting checklist for researchers conducting human baseline studies. We hope our work can advance more rigorous AI evaluation practices that can better serve both the research community and policymakers.[1]

## 1 INTRODUCTION

Artificial intelligence (AI) systems, and foundation models in particular, have increasingly achieved superior performance on benchmarks in natural language understanding, general reasoning, coding, and other domains (Maslej et al., 2024). These results are frequently compared to *human baselines*—reference sets of metrics intended to represent human performance on specific tasks—which has led to claims about models' "super-human" performance (Bikkasani, 2024).

Human baselines are crucial for evaluating AI systems and for understanding AI's societal impacts. For the machine learning (ML) research community, human baselines help improve benchmarks, provide context for interpreting system performance, and demonstrate concurrent validity (Hardy et al., 2024; Bowman & Dahl, 2021). For downstream users, comparisons to human performance may inform decisions about AI adoption (cf. Luo et al. 2019). And for policymakers, human baselines facilitate risk assessments (OSTP, 2022; NIST, 2023; Goemans et al., 2024; US AISI & UK AISI, 2024) and predictions of AI's economic impacts (Hatzius et al., 2023; Shrier et al., 2023). Valid and reliable human baselines thus contribute greatly to the operational value of AI evaluations.

However, despite widespread recognition in the ML community about the importance of human baselines (Reuel et al., 2024; Ibrahim et al., 2024; Tedeschi et al., 2023; Nangia & Bowman, 2019; Bender, 2015), existing human baselines are neither sufficiently rigorous nor sufficiently transparent to enable reliable claims about (the magnitude of) differences between human and AI performance. For instance, human baselines in many evaluations have small or biased samples (Liao et al., 2021;

---

[*]Equal contribution. Correspondence to: `hi@kevinlwei.com`
[1]Data is available at: `https://github.com/kevinlwei/human-baselines`

McIntosh et al., 2024), apply different instruments than those used in AI evaluation (Tedeschi et al., 2023), or fail to control for confounding variables (Cowley et al., 2022). In addition, published evaluations commonly omit study details necessary for assessing baseline validity, such as how participants were recruited or how questions were administered (Section 4.5). Measurement theory, a methodological field in the social sciences concerned with quantifying complex concepts, addresses analogous issues in human studies (Bandalos, 2018) and can inform best practices in human baselines.

**Our position is that human baselines in evaluations of foundation models must be more rigorous and more transparent.** Building from measurement theory, we propose a framework for assessing human baselines. Using our framework to systematically review 113 published human baselines, we find substantial shortcomings in existing human baselining methods. We hope that our framework can support researchers in developing human baselines that are more interpretable and valuable to the ML community, downstream users, and policymakers.

In defending our position, we acknowledge that there are challenges with building rigorous human baselines, such as the expense of high-quality baseline data, the evolving landscape of AI evaluations, and differences in cognition and interaction modes between humans and AI systems. *Given these complexities, our framework is not intended to be a one-size-fits-all prescription for all AI evaluations—rather, our work proposes a reference set of operational considerations to inform researchers in designing and implementing human baselines.*

We proceed as follows: Section 2 presents related work and background. Section 3 describes our methodology (full details in Appendix B). Section 4 presents our framework and the results of our systematic review, which examines the entire lifecycle of human baselining including baseline(r) design, recruitment, implementation, analysis, and documentation. Section 5 discusses results and limitations. Section 6 concludes.

## 2 BACKGROUND

*Measurement theory* is the discipline devoted to quantifying complex or unobservable *concepts* through the use of observable indicators, or *measurements* (Goertz, 2020). Since concepts are often multidimensional or impossible to measure directly, researchers usually aggregate multiple measurements and rely on proxies for quantities of interest. *Intelligence*, for instance, has sometimes been measured by aggregating multiple different cognitive tests (Deary, 2012). In the social sciences, measurement theory has also been applied to concepts such as fairness (Patty & Penn, 2019), emotion (Reisenzein & Junge, 2024), culture (Mohr & Ghaziani, 2014), personality (Drasgow et al., 2009), and language (Sassoon, 2010). Measurement theory helps build indicators for these concepts that satisfy criteria of *validity* (yielding results that support intended interpretations of measurements) and *reliability* (yielding consistent results across multiple measurements) (Bandalos, 2018).

There has been growing recognition in the AI research community that AI evaluation can draw valuable lessons from measurement theory and the social sciences (Chang et al., 2024b; Wallach et al., 2024; Eckman et al., 2025; Chouldechova et al., 2024; Blodgett et al., 2024; Xiao et al., 2023; Zhou et al., 2022; Zhao et al., 2025; Wang et al., 2023; Liao & Xiao, 2023). Like measurement theory, AI evaluation has been concerned with estimating concepts such as intelligence, fairness, emotion, and culture—though in AI models rather than in humans (Chang et al., 2024b). Research in ML has focused in particular on applying measurement theory to performance metrics (Subramonian et al., 2023; Flach, 2019) and fairness metrics (Jacobs et al., 2020; Grote, 2024; Blodgett et al., 2021). Additionally, measurement theory provides frameworks for making comparisons between (human) populations—analogous to the problem of comparing human and AI performance, which is often addressed using human baselines in AI evaluations.

We draw on measurement theory to examine human baselining in evaluations of *foundation models* (Bommasani et al., 2022), which pose unique evaluation challenges (Liao & Xiao, 2023). Applying measurement theory to the foundation model context is particularly appropriate as foundation models are exhibiting increasingly general, multidimensional capabilities (Zhong et al., 2024) and beginning to interact with the same interfaces as human users (Anthropic, 2024b; Chan et al., 2025). Specifically, we adopt the approach of Zhao et al. (2025) in drawing on measurement theory to validate the data generation process in human baselines—that is, we are particularly concerned with the validity and reliability of baselining *methods* rather than the trustworthiness of any particular baseline data.

Analysis of the full pipeline of human baselining methods in the foundation model context is limited. Prior work has critiqued human annotation processes (Tedeschi et al., 2023) and offered high-level principles for human baselining (Cowley et al., 2022). Building on this literature, we provide a comprehensive discussion of operational-level methodological considerations for human baselining. We also fill a gap in the literature by systematically reviewing human baselines in foundation model evaluations, allowing us to identify shortcomings of and opportunities for improvement in existing methods for human baselining.

## 3 METHODOLOGY

We used a two-stage approach to develop our position, adapted from Zhao et al. (2025) and Reuel et al. (2024). First, we conducted a meta-review (review of reviews) of the measurement theory literature to construct a checklist of practices for baselining (Appendix A). Using purposive sampling and backwards snowballing, we identified 29 articles from the social sciences (psychology, economics, political science, education) and AI evaluation. The checklist was initially compiled after reviewing these articles, then refined through internal discussion and expert validation.

Second, we conducted a systematic review (see Page et al. 2021) of the AI evaluations literature to identify gaps in existing human baselining methods. From academic publications and gray literature, we identified 113 human baselines in foundation model evaluations. Inclusion criteria consisted of whether the article contained 1) an original human baseline, 2) an evaluation of a foundation model, and 3) both a human baseline-related keyword ("human baseline*", "expert baseline*", "human performance baseline*") and an AI evaluation-related keyword ("AI evaluation*", "ML benchmark*", etc.). Articles were then manually coded per the checklist from our meta-review (Appendix A), with the codebook iteratively refined during the coding process. Full methodological details are in Appendix B.

Figure 1: Our synthesized framework for human baselining. The full checklist can be found in Appendix A.

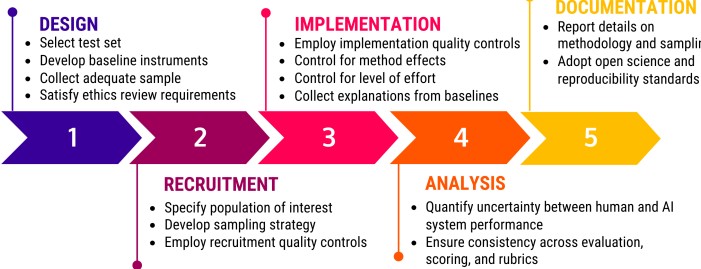

## 4 A FRAMEWORK FOR RIGOROUS AND TRANSPARENT HUMAN BASELINES

We organize our discussion by delineating five stages of the baselining process, as adapted from Reuel et al. (2024): design, recruitment, implementation, analysis, and documentation. For simplicity, we also synthesize our checklist (Appendix A) into an operational framework of key factors to consider at each stage of human baselining. Neither our checklist nor our framework is intended to provide hard recommendations but rather to inform researchers in developing baselines and reporting results. Our framework is presented below and summarized in Figure 1.

### 4.1 BASELINE DESIGN

Baseline design is the initial stage of human baseline development, at which researchers define baselines' purpose, scope, concepts, evaluation items, and metrics. (Reuel et al., 2024). We examine four considerations for baseline design.

**Selecting the test set for a human baseline.** Robust comparisons of human vs. AI performance require contrasting performance on the same test set. Because the cost of human baselines can make

baselining on a large number of items infeasible, researchers often construct baselines using subsets of the test sets used for AI evaluation. Baseline validity thus depends on the sampling strategy used to create the human baseline test set. Of the baselines in our review, 51% (58 of 113) used different test sets for AI evaluation and human baselines when presenting results.

Baseline test sets (where specified) were most commonly created using simple random (19% (21)), stratified (13% (15)), or purposeful sampling strategies (3% (3)). Simple random sampling from the broader evaluation dataset may be sufficient to ensure representativeness of the baseline test set, assuming the test set is sufficiently large (see Liao et al. 2021). Stratified sampling may be preferred where the baseline test set is relatively small, or where the test set must preserve important properties of the evaluation dataset such as data source (e.g., Xiang et al. (2023)), question difficulty (Tedeschi et al., 2023), or other relevant dimensions (Cowley et al. 2022; e.g., Liu et al. 2024b, Bai et al. 2024).

When reporting human baseline results, researchers should clearly indicate where human baseline test sets differ from AI evaluation test sets. To directly compare AI results with human baselines, researchers can also report AI performance on only the human baseline test set.

**Using an iterative processes to develop baseline instruments.** Iterative processes repeatedly test and refine the instruments (e.g., forms, surveys, prompts) used to measure constructs by applying multiple rounds of validation, feedback, and refinement of baseline instruments before data collection. 34% (38) of baselines reviewed used iterative methods during baseline design.

The feedback loops created by iteration can support construct validity (Rosellini & Brown, 2021) while ensuring clarity and consistent interpretation of instruments (Cheng et al., 2024; Cowley et al., 2022). In the social sciences, iterative processes are the gold standard in the social sciences for collecting annotations (Cheng et al., 2024), running surveys (Groves et al., 2011), and building clinical questionnaires (Rosellini & Brown, 2021). The ML community has also recognized the importance of iteration, such as when optimizing AI prompts (Hewing & Leinhos, 2024; Gao et al., 2025). Researchers often validate items in evaluation *datasets* (e.g., Nangia et al. 2021; Rein et al. 2024) but less frequently validate baseline *instruments*. An evaluation that optimizes AI prompts and validates evaluation items, but that does not validate baselining instruments, may unfairly disadvantage humans and thereby discount baseline validity.

Iteration does add complexity to the baselining process. However, it is not necessarily costly: large pilot studies and focus groups may be out of reach to budget-constrained researchers, but small-scale pre-tests or expert validation can still help improve baselining artifacts (Groves et al., 2011; Zickar, 2020).

**Collecting an adequately sized sample of baseliners.**[2] Baselines that are underpowered because of small sample sizes are unreliable because they cannot robustly capture the underlying distribution of human performance (Cao et al., 2024). Power analyses can help determine an appropriate sample size for human baselines, given significance levels and pre-specified minimum detectable effect sizes in the outcome metric of interest (Cohen, 2013). The importance of statistical power in ML benchmarks has been noted in prior work (Bowman & Dahl, 2021; Grosse-Holz & Jorgensen, 2024), but only 2% (2) of the human baselines we reviewed reported conducting power analyses.

If sample sizes are fixed (e.g., due to cost constraints), researchers can nevertheless calculate and report the required sample size to reliably detect practically important effects, which supports interpretation of evaluation results. Understanding the ability of human baselines to detect performance differences may be especially important to users and policymakers, who may demand added rigor and certainty in evaluation results to inform decision-making (Paskov et al., 2024). In general, considerations around statistical power reflect broader shortcomings in using statistical methods in AI evaluation, which we discuss further in Section 4.4.

**Satisfying ethics review requirements for human subjects research.** Ethics review protects human research participants (Page & Nyeboer, 2017), and it is legally required in many jurisdictions, including in the U.S. (U.S. Department of Homeland Security et al., 2017). Significantly, only 13% (15) of the articles we examined reported compliance with or formal exemption from ethics review requirements, near the same order of magnitude as the 2% found by Kaushik et al. (2024).

---

[2]"Sample" in this context refers to the subset of humans in the baseline who are drawn from an underlying population.

Reporting compliance with ethics requirements is best practice in many fields, e.g., medicine (ICMJE, 2025). Failure to report compliance in an article does not indicate failure in compliance, and some evaluations may be exempt from review (Kaushik et al., 2024). However, transparency around research ethics becomes more important as public interest in AI increases, and protection of research participants can also be critical for evaluations implicating, for example, deception, misinformation, and psychological impacts.

## 4.2 BASELINER RECRUITMENT

Baseliner recruitment is the stage at which human baseliners—the humans who respond to evaluation items—are found and are engaged to participate in a baseline. We examine three considerations for baseliner recruitment.

**Specifying a human population of interest.** Specifying a population of interest—the group of humans for whom a baseline is intended to be representative—is a cornerstone of valid human subjects research. Prior work has noted that AI evaluations rarely specify populations of interest (Subramonian et al., 2023), which is in line our review: only 31% (35 of 113) baselines explicitly or implicitly defined a population of interest along at least one axis (i.e., a population beyond "humans").

Populations of interest can be specified through axes such as geographic location, demographic characteristics (e.g., age, gender, socioeconomic status), language, cultural background, education, or domain expertise. A human baseline may seek to measure the performance of, for instance, a population of medical or legal professionals (e.g., Blinov et al. 2022; Hijazi et al. 2024). In published AI evaluations that defined a population of interest, the most commonly reported characteristics were education (20% (23 of 113)), expertise (19% (22)), language (18% (20)), and age (18% (20)). How to scope the population of interest for any given baseline will depend on the evaluation's research questions, context, and intended use.

**Developing a sampling/recruitment strategy for selecting baseliners.** Sampling strategy is the process by which humans are selected to participate in the baseline, such as convenience sampling or random sampling. Sampling strategy directly informs how representative the selected baseliners are of the population of interest, and representativeness is essential for external validity because it determines whether baseliners' results can be generalized to a larger population of humans (Findley et al., 2021; Stantcheva, 2023; Berinsky, 2017). Of the baselines in our review, 32% (36) used a convenience sample, 31% (35) recruited from crowdsourcing platforms, and the remainder did not report their sampling strategies.

Both convenience sampling and crowdsourced samples present sampling challenges that are rarely addressed in the AI evaluation literature. Convenience samples, such as samples of undergraduates or of an article's authors, are highly susceptible to significant biases that reduce their applicability to broader populations (cf. Mihalcea et al. 2024; Diaz & Smith 2024). Recruitment of baseliners from crowdsourcing platforms, such as Amazon Mechanical Turk (MTurk) or Prolific, also poses challenges to representativeness. Prior research has shown that MTurk workers tend to be younger, more educated, more politically liberal, and more engaged online compared to the general population (Sheehan, 2018; Shaw & Hargittai, 2021; Stantcheva, 2023). Crowdsourced baselines may thus fail to represent performance of humans who do not reflect those demographics. Even if crowdsourced samples are sufficiently large from the perspective of statistical power, results may still be biased if the sample is insufficiently representative of the underlying population of interest (Bradley et al., 2021).

Researchers designing human baselines should consider methodological adjustments to ensure baseliner representativeness. For instance, researchers could consider stratified sampling to improve representativeness along specific dimensions (Groves et al., 2011). Post hoc adjustments such as weighting may partially mitigate selection bias in non-representative samples (Solon et al., 2015). Researchers may also explore non-probability sampling approaches that aim to enhance representativeness (Couper, 2017).

At a minimum, researchers should clearly report the sampling strategy used, acknowledge the limitations of non-representative samples, discuss which populations results may apply to, and discuss implications for the validity of baseline results.

**Employing quality controls for baseliner recruitment.** Quality control (QC) mechanisms at the recruitment stage improve data quality by selecting for baseliners who can generate high-caliber data. Using inclusion/exclusion criteria during recruitment is considered best practice in survey research (Stantcheva, 2023) and ensures that baseliners meet appropriate evaluation criteria. QC mechanisms can include pre-testing baseliners for task-specific knowledge or general ability (see, e.g., Nangia et al. 2021) or filtering crowdsourced workers via screening questions or platform quality scores (Lu et al., 2022a).

Depending on the research question, baseliners' domain expertise may be particularly important because expert baseliners often provide higher-quality data than non-experts (Cheng et al., 2024; Liao et al., 2021). Specialized (expert) human baselines are specifically needed to enable AI evaluations that compare AI performance with the ceiling of possible human performance and that explore the possibility of "super-human" performance (e.g., Glazer et al. 2024).

Considerations for QC in recruitment include the cost and feasibility of recruiting (expert) baseliners, whether evaluations items require different QC criteria or expertise (cf. Weidinger et al. 2024), and how to establish criteria for assessing baseliners (e.g., assessing domain expertise in highly-specialized evaluations). Researchers may also wish to exclude baseliners who have been previously exposed to evaluation items to prevent data contamination, analogous to AI train/test contamination.

### 4.3 BASELINE IMPLEMENTATION

Baseline implementation is the stage at which human baseline data is collected—e.g., through surveys or crowdwork platforms. We examine four considerations for baseline implementation.

**Employing quality controls in baseline implementation.** Quality control mechanisms at the implementation stage improve data quality by filtering out unreliable baseline responses. As with the recruitment stage, QC during implementation is considered best practice in survey research; mechanisms include checks for attention, consistency, response pattern, outliers, and time to completion (Stantcheva, 2023; Lebrun et al., 2024). Some research has also demonstrated that attention checks may improve the representativeness of crowdwork samples (Qureshi et al., 2022). Of the baselines in our review, 23% (26 of 113) reported performing QC at the implementation stage, most often using attention checks and honeypot questions.

One issue of increasing importance is the inappropriate usage of AI tools by crowdworkers, which was directly raised as a concern in one article we reviewed (Sprague et al., 2023). Empirical work has suggested that a substantial percentage of MTurk workers have used AI to complete tasks (Veselovsky et al., 2023b; Traylor, 2025), which can decrease data quality (Lebrun et al., 2024) and baseline validity. Unintentional usage of AI tools may also occur as AI adoption increases such as via AI-generated Google Search summaries. QC mechanisms to prevent AI usage may be beneficial for crowdsourced baselines, e.g., explicitly asking participants not to use LLMs (Veselovsky et al., 2023a), employing technical restrictions such as preventing copy/pasting (Veselovsky et al., 2023a), using non-standard interface elements (Gureckis, 2021), or using comprehension and manipulation checks (Frank et al., 2025). That said, AI use is not always undesirable, such as in domains where AI usage is expected, in which evaluations can compare AI capabilities with baselines of AI-augmented human capabilities (e.g., Wijk et al. 2024).

**Controlling for method effects.** Method effects are variations in item response attributable to data collection methods rather than to differences in underlying response distributions, and they can reduce evaluations' internal validity (Davidov et al., 2014). Our review found significant discrepancies in data collection methods between humans and AI models: 35% (40) of baselines displayed UI differences between human baselining and AI evaluation, 19% (22) displayed differences in instructions or prompts, and 7% (9) displayed differences in tool access. Note that our focus was on method effects between human and AI responses, but method effects can also occur between human baseliners (e.g., if baselines are collected from multiple platforms).

Method effects are well-documented in the social sciences, particularly in psychology and in survey methodology. Empirical research has found effects due to the mode of survey administration (Vannieuwenhuyze et al., 2010; Shin et al., 2012), question order (Engel et al., 2014), fatigue from survey length (Stantcheva, 2023), example responses provided (Eckman et al., 2025; Lu et al., 2022b), interface design (Sanchez, 1992), and question wording (Wu & Quinn, 2017; Dafoe et al., 2018).

Measurement theory offers some guidance for addressing method effects in humans. For instance, randomization of non-critical methodological details can reduce some effects (e.g., reducing order effects by randomizing question order). Fatigue can also be addressed by shortening survey length, encouraging breaks or enforcing time limits, and implementing attention checks. However, not all such guidance is applicable to AI systems.

Some method effects in AI evaluation are currently unavoidable due to differences between human and AI cognition. For instance, many AI evaluations restart the context window for each run, but it may be unrealistic to demand that baseliners are only administered one item per sitting; only 27% (30) of reviewed baselines reported instrument length, of which most reported instruments (24) were longer than one item. Similarly, although both AI systems and humans are known to be sensitive to item wording, they are sensitive in different ways (Tjuatja et al., 2024), suggesting that simply using the same data collection artifacts for humans and for AI systems may not prevent method effects.

Overall, significant additional research is needed to understand how method effects differ between humans and AI systems (and between AI systems), as well as how AI evaluations can adjust for these differences so as not to unfairly advantage humans or AI models in the evaluation process (Cowley et al., 2022; Tedeschi et al., 2023). In the absence of clear evidence, researchers can contribute to this area of research by clearly documenting evaluation methodologies.

**Controlling for level of effort.** Both humans' and AI systems' level of effort in responding to items can affect evaluation results, and baseliner effort can in turn be affected by training, compensation, and other factors (Tedeschi et al., 2023). Training could include tutorials, response guides, or example items; compensation structures can also affect baseline data quality (Grosse-Holz & Jorgensen, 2024) and can vary by, e.g., payment by hour vs. per task or by performance bonuses. Our review found that 22% (25) of baselines provided training to baseliners, and 39% (44) paid baseliners, with 6% (7) providing performance bonuses.

Accounting for level of effort also raises design questions about the relevant experimental unit of interest, the choice of which affects evaluations' external validity (Jackson & Cox, 2013). Most AI evaluations take humans or AI systems as the experimental unit, but some comparisons may necessitate more granularity. For instance, Wijk et al. (2024) compares performance after two human labor-hours and after two AI labor-hours. Properly scoping experimental units could make evaluations more valuable for understanding AI's broader societal effects, e.g., by enabling comparisons of labor efficiency.

**Collecting explanations from baseliners.** In some instances, researchers may find value in explanations of why baseliners chose particular responses. Collecting explanations is generally a best practice in survey research, as it can help surface new insights about data (Lu et al., 2022a). For AI evaluations, explanations can reveal qualitative differences between human and AI performance, explain performance gaps, and surface validity issues. Explanations may also be used for quality control, validation, and understanding the thought processes behind item responses (Lu et al., 2022a; Tedeschi et al., 2023), which may lead to improvements in questions or instrumentation. Only 10% (11) of the baselines we reviewed collected explanations from baseliners, though this finding is unsurprising since collecting explanations may increase the cost of human baselines, and it is unclear whether explanations are necessary in many baselines.

## 4.4 Baseline Analysis

Baseline analysis is the stage after data collection at which human baseline data is inspected and compared to AI results. We examine two considerations at the analysis stage.

**Quantifying uncertainty in human vs. AI performance differences.** Reporting measurements of uncertainty or applying statistical tests is necessary to rigorously assess whether measurements of performance truly reflect underlying performance distributions, as well as to interpreting evaluation results (Agarwal et al., 2022; Steinbach et al., 2022). The ML community has historically recognized these norms (e.g., Dietterich 1998; Bouckaert & Frank 2004), but many recent evaluations of large AI models have not met standards of statistical rigor (Biderman & Scheirer, 2020; Agarwal et al., 2022; Welty et al., 2019; Paskov et al., 2024). Similarly, our review finds that only 26% (29 of 113) of evaluations provided interval or distribution estimates, and only 9% (10) performed statistical tests of any type.

Lack of statistical testing is sometimes understandable given small sample sizes and other evaluation limitations (cf. Bouthillier et al. 2021). Reporting interval estimates, however, has become increasingly accessible with increased guidance (e.g., Miller 2024) and support in major evaluation frameworks (e.g., UK AISI 2025). The ML research community could also draw on established statistical methods for analyzing small samples (Neuhäuser & Ruxton, 2024; Hoyle, 1999; Schoot & Miočević, 2020). Finally, in line with recent commentary in statistics, researchers should consider reporting results of statistical tests ($p$-values) as one component of evidence used to judge the evaluation results, rather than as screens for statistical significance (McShane et al., 2019; Gelman & Stern, 2006).

**Using consistent evaluation metrics, scoring methods, and rubrics across human and AI evaluation.** Often, comparisons between AI and human baseline results are fair only when the metrics for comparison are equivalent across samples. For instance, human baseline metrics are sometimes calculated inconsistently across items, complicating baseline interpretation (Tedeschi et al., 2023). In our review, 89% (100) of baselines used the same evaluation metrics across human and AI results, but only 59% (67) and 63% (71) used the same scoring rubric and scoring method. Most commonly, researchers used majority vote for human but not for AI samples. Although these comparisons are not always inappropriate, researchers should consider adding clarifying language when reporting results, e.g., "AI evaluation metrics fell below majority-vote human performance" or "model results on each item exceeded the maximum performance across ten human baseliners."

## 4.5 BASELINE DOCUMENTATION

Baseline documentation is the provision of evaluation tasks, datasets, metrics, and experimental materials and resources (Reuel et al., 2024). We examine two considerations for baseline documentation.

**Reporting key details about baselining methodology and baseliners.** Documentation includes reporting information about baseliners, baselining procedures, and baseline paradata. These details can significantly affect how results are contextualized and interpreted—especially with respect to their validity—and reporting can build collective confidence in published results (Liao et al., 2021).

We believe that absent compelling reasons for confidentiality, researchers should strongly consider documenting most items in our checklist related to baseline(r) design, recruitment, implementation, and analysis (Appendix A). Researchers should also consider reporting baseliner demographics, paradata, and other study information. Baseline demographics can enable assessments of baseliner sample representativeness and reliability; paradata such as items' time to completion can offer insights into latent variables like cognitive effort (Cai et al., 2016) and into data quality, which is often correlated with response times (Traylor, 2025). Of the baselines in our review, all failed to report at least some items on our checklist, only 20% (23 of 113) provided detailed baseliner demographics, and only 24% (27) included paradata such as response times.

**Adopting best practices for open science and reproducibility/replicability.** Releasing human baseline data, experimental materials (e.g., forms, custom UIs), and analysis code in accessible repositories (e.g., GitHub, OSF) can facilitate research validation and reproduction/replication (Semmelrock et al., 2024; Stodden & Miguez, 2014). These open science practices also facilitate reuse of human baseline data in subsequent evaluations by other researchers, which can in turn foster more efficient use of resources within the ML community. Concerns around open science and replicability are not new in ML (Kapoor et al., 2024; Pineau et al., 2021), and our review found that most baselines (80% (90)) did not publicly release human baseline responses, experimental materials (57% (64)), and code for analyzing human baselines (52% (59)).

## 5 DISCUSSION

In this section, we discuss three additional considerations for human baselines and address the limitations of our study. We discuss and respond to alternative views in Appendix D.

**First, human baselines are not appropriate for all AI evaluations**. Most prominently, human baselines are not meaningful for evaluations of AI tasks without human equivalents (Barnett & Thiergart, 2024). Examples include AI control evaluations, which measure an AI system's ability to monitor a more advanced AI system (Greenblatt et al., 2024), and autonomous self-replication

evaluations, which measure an AI agent's ability to create copies of itself (Pan et al., 2024). In contrast, human baselines can be valuable for evaluations that measure AI performance in domains with human equivalents, including but not limited to many question-answer benchmarks and task-based agent evaluations (e.g. Wijk et al. 2024).

**Second, human baselines may also be constructed from secondary sources.** Our position paper focuses on primary data collection methods in human baselining, but human performance metrics can also be derived from real-world data or pre-existing datasets. For instance, the Massive Multitask Language Understanding dataset uses the $95^{th}$ percentile of human standardized test scores as a point of comparison with AI results (Hendrycks et al., 2021a); other studies (e.g., Hua et al. 2024) use human subjects data from previous work (Lewis et al., 2017). Re-use of human baselines highlights the need for transparency and documentation of baselining methods: authors should assume their datasets may be re-used by other researchers, who require significant methodological detail to design effective evaluations and draw meaningful conclusions from results. Secondary data is also subject to many other limitations (see Appendix D).

**Third, human baselines can vary over time and as technology advances.** Human capabilities are known to change over time (Trahan et al., 2014), and the half-life of AI-augmented human baselines (e.g., Wijk et al. 2024) may be particularly short due to the rate of progress in AI. These trends suggest that human baselines should be understood to represent measurements at specific points in time, and researchers should tread carefully when making comparisons to older human baselines. In this vein, the ML community can consider implementing regularly updated "living" baselines, analogous to how public opinion polls are regularly repeated to track variation over time. Open science practices would enhance replicability and resource efficiency for such living baselines.

**Finally, we acknowledge several limitations to our work.** Our methodology has generally followed best practices for systematic literature reviews, but our meta-review sample was collected purposively and could be biased as a result (see discussion in Appendix B.1). Our scope is limited to methodological considerations specific to human baselines, so we do not discuss many important aspects of AI evaluation methodology such as construct validity (Strauss & Smith, 2009). We also restricted our scope to foundation model evaluations; although we believe much of our framework is applicable to the broader research community, human baselines for evaluating other AI models may raise new methodological questions. Finally, future research can examine applications of measurement theory to human evaluation and human-AI interaction studies (e.g. human uplift), which are not explored in this position paper.

# 6 CONCLUSION

In this position paper, we argued that human baselines in foundation model evaluations should be more rigorous and transparent. Systematically reviewing 113 published evaluations, we find that many baselines lack methodological rigor across the gamut of the baselining process, from design (e.g., test set selection) to documentation (e.g., lack of study details). We provide an instructional framework for human baselining based on measurement theory. Our approach fosters validity and reliability in human baselines, enabling meaningful comparisons of human vs. AI performance and promoting research transparency. We hope that our work can guide researchers in improving baselining methods and evaluating AI systems.

## IMPACT STATEMENT

This paper presents work whose goal is to advance the field of machine learning, specifically with regards to improving the quality of methods used to create and analyze human baselines in AI evaluation. We hope that by discussing methodological considerations in human baselining—and by highlighting shortcomings in existing baselining methods—our work will lead to rigorous AI evaluations that can be useful to not just the research community but also to users of AI systems and policymakers. We discuss broader implications of human baselines in Section 1, and we do not anticipate any particular negative impacts associated with our work.

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

## A    APPENDIX: FULL CHECKLIST

Our checklist is presented in full below, updated with slight modifications and reorganization from the version used in our coding process. Our hope is that this checklist can guide and inform researchers in building human baselines and in reporting baseline results.

Note that the following changes were made during our coding process:

- All items were open text fields unless explicitly indicated otherwise below.
- For questions on a scale of "Yes", "Partial", "No", "Unknown/Unreported", or "N/A":
  - "Yes" and "No" options were selected only if the relevant checklist item was explicitly noted in an article's main text, supplementary material/appendices, or GitHub codebase.
  - 'Partial" was selected where articles did not fully satisfy the item criterion, e.g., satisfying the criterion for some but not all of the baseline items. "Partial" labels were "rounded" up to "Yes" labels unless otherwise specified below.
  - "Unknown/Unreported": see below.
  - "N/A" was selected where the item did not apply to the baseline at hand.
- For all questions, including items with open text fields: coders indicated "Unknown/Unreported" where items were not reported or where coders were not able to determine the response based on an article's main text, supplementary material/appendices, or GitHub codebase.
  - For select items, "Unknown/Unreported" labels were resolved to default values, which are indicated below in underline and with a "(Default)" label. Default responses are selected based on our understanding of common practices in AI evaluation, and we attempt to be liberal in terms of assuming rigor in the baseline where there are is no consensus in the literature on common practices.
  - For items without default responses, "Unknown/Unreported" labels were not adjusted.

### A.0    PAPER INFORMATION

0.1 **Paper Title**

0.2 **Paper Link**

0.3 **Publication Year**

0.4 **Publication Venue**

0.5 **Type of Eval**
*Select all that apply*

- Knowledge
- Capabilities
- Propensity
- Agent

0.6 **Mode of Eval**
*Select all that apply*

- Text
- Visual (photo/video)
- Audio
- Other

0.7 **Language of Eval**
*Select all that apply from list*

0.8 **Evaluation Dataset Size**: What is the total number of items in the evaluation dataset?

0.9 **AI Test Set Size**: What is the number of items that the AI evaluation is run on? (Default same as Q0.8)

0.10 **AI Samples per Item**: What is the number of AI responses ("samples" or "runs") that is collected for each item? (Default 1)

## A.1 BASELINE DESIGN

1.1 **Number of Baseliners**: How many baseliners were there total?

1.2 **Baseline Test Set Size**: What is the number of items that the human baseline is run on? (i.e., how many of the questions do the baseliners collectively answer?) (Default same as Q0.9)

   1.2.1 **Baseline Test Set Sampling Strategy**: If the baseline is only run on a sample of the total dataset: what is the sampling strategy behind how the items were selected? E.g., simple random sampling, stratified sampling, etc.

1.3 **Baseline Samples per Item**: What was the number of human baseliner responses that is collected for each item? (Default $Q1.1 * Q1.4 / Q1.2$, or 1 if Q1.1 or Q1.4 unreported)

1.4 **Items per Baseliner**: What is the number of items that each baseliner responded to?

1.5 **Explicit Human/AI Adjustment**: Does the eval/baseline instructions and items account for both humans and AI models completing the evals items (questions/tasks)? E.g., do the authors of the eval explicitly state that the eval is designed so as not to advantage either humans or AI models?
*Select one of: "Yes", "Partial", "No" (Default), "Unknown/Unreported", or "N/A"*

1.6 **Iterative Design**: Was the experimental setup of the baseline iteratively designed with participatory methods? E.g., was there a pilot study, expert validation of the items, etc.?
*Select one of: "Yes", "Partial", "No", "Unknown/Unreported", or "N/A"*

1.7 **Amount of Effort**: Does the baseline control for the amount of effort by human baseliners and AIs? E.g., in terms of cost, time, etc.
*Select one of: "Yes", "Partial", "No", "Unknown/Unreported", or "N/A"*

1.8 **Power Analysis**: Did the authors conduct power analysis in order to determine baseline size?
*Select one of: "Yes", "Partial", "No" (Default), "Unknown/Unreported", or "N/A"*

   1.8.1 **Minimum Detectable Effect Size**: if yes, what is the minimum detectable effect size and power?

1.9 **Ethics Review**: Was the study approved or exempted by an IRB, or did it undergo other ethics review?
*Select one of: "Yes", "Partial", "No", "Unknown/Unreported", or "N/A"*

1.10 **Pre-Registration**: Was the baseline/eval design pre-registered? I.e., a plan detailing the experimental setup that is publicly registered online before running the experiment (e.g., on OSF, COS, etc.)
*Select one of: "Yes", "Partial", "No" (Default), "Unknown/Unreported", or "N/A"*

## A.2 BASELINER RECRUITMENT

2.1 **Population of Interest Identification**: Does the reporting identify human populations for which these results may be valid, i.e., a human population of interest?
*Select one of: "Yes", "Partial", "No" (Default), "Unknown/Unreported", or "N/A"*

   2.1.1 **Population of Interest Identification Criteria**: Which of the following factors were used to scope the target human population of interest?
*Select all that apply*
- Expertise
- Education
- Language
- Gender/sex
- Race
- Socioeconomic status
- Age
- Disabilities/impairments
- Political orientation
- Digital literacy (Prior experience with computers)
- AI literacy (Prior experience with AI tools)
- Baseline experience: Prior experience with AI evals/doing human baselines
- Other (specify)

2.2 **Baseliner Sampling Strategy**: How were the human baseliners recruited?
*Select one of the below*

- Crowdsource
- Convenience sample
- Simple random sample
- Stratified random sample
- Other (specify)
- Unknown/unreported

2.3 **Quality Control in Recruitment**: Were human baseliners pre-qualified or excluded during the recruitment process for any reason?
*Select one of: "Yes", "Partial", "No", "Unknown/Unreported", or "N/A"*

2.3.1 **Quality Control Criteria for Baseliners**: If yes: please describe the inclusion/exclusion criteria for human baseliners (e.g., pre-tests, expert judgements/filtering, quality scores or ratings on crowdwork platforms, number of tasks completed on crowdwork platforms). Data quality checks that occurred after baseliners were recruited should be reported in the implementation section (e.g., attention checks in a survey).

2.3.2 **Recruitment Exclusion Rate**: If yes: how many baseliners were excluded from the final baseline based on these criteria?

2.4 **Author Baseliners**: Did the authors or members of the research team also serve as human baseliners?
*Select one of: "Yes", "Partial", "No" (Default), "Unknown/Unreported", or "N/A"*

2.5 **Baseliner Train/Test Contamination**: Did the recruitment process exclude baseliners who had been exposed to the eval questions previously?
*Select one of: "Yes", "Partial", "No" (Default), "Unknown/Unreported", or "N/A"*

2.6 **Baseliner Training**: Did the human baseliners receive training for the baseline? Training should be distinct from the reported data, e.g., a tutorial completed before answering baseline questions
*Select one of: "Yes", "Partial", "No" (Default), "Unknown/Unreported", or "N/A"*

2.6.1 **Baseliner Training Type**: If yes: describe the type of training received (e.g., tutorial, shown examples, etc.)

2.6.2 **Baseliner Training Compensation**: If yes: were the baseliners compensated for the training?
*Select one of: "Yes", "Partial", "No", "Unknown/Unreported", or "N/A"*

2.6.2.1 **Baseliner Training Compensation Amount**: If yes: list the compensation per baseliner (preferably $ / hour, otherwise total $ amount if stated)

2.7 **Baseliner Testing Compensation**: Were the human baseliners compensated for completing the baseline?
*Select one of: "Yes" (Default if Q2.2 is "Crowdsource"), "Partial", "No", "Unknown/Unreported", or "N/A"*

2.7.1 **Baseliner Testing Compensation Amount**: If yes: how much was compensation? (preferably $ / hour, otherwise total $ amount if stated)

2.7.2 **Baseliner Testing Performance Bonus**: If yes: was a performance bonus offered to baseliners?
*Select one of: "Yes" (Default if Q2.2 is "Crowdsource"), "Partial", "No", "Unknown/Unreported", or "N/A"*

2.7.2.1 **Baseliner Testing Performance Bonus Amount**: If yes: how much was the performance bonus, and how was it determined?

2.7.3 **Baseliner Testing Compensation Structure**: If yes: were compensation rates and structures constant across baseliners? E.g., respond no if baseliners were paid differently according to expertise.
*Select one of: "Yes", "Partial", "No", "Unknown/Unreported", or "N/A"*

2.7.3.1 **Baseliner Testing Compensation Structure Details**: If not compensated equally: how were compensation amounts determined?

A.3   BASELINE IMPLEMENTATION

3.1 **Instrument Length**: How many items did the human baseliners complete in a single sitting/session? I.e., what is the length of the baseliner "context window" in units of items?

   3.1.1 **Item Randomization**: If not 1: was the order of the questions randomized?

3.2 **Quality Control in Implementation**:   Were quality checks implemented or data cleaned/excluded during the data collection process (i.e., after baseliners were recruited)? E.g., were there any exclusion criteria for baseliner responses due to data quality such as attention check questions, honeypot questions, filtering out responders who completed the eval too quickly, screen recording, etc.
   *Select one of: "Yes", "Partial", "No", "Unknown/Unreported", or "N/A"*

   3.2.1 **Quality Control in Implementation Criteria**: If yes: what factors were used to determine data quality or to exclude low-quality data?

   3.2.2 **Implementation Exclusion Rate**: If yes: how many samples were excluded from the final baseline based on these criteria?

3.3 **UI Equivalence**: Did the human baseliners and AIs have access to the same UI for each item?
   *Select one of: "Yes", "Partial", "No" (Default), "Unknown/Unreported", or "N/A"*

   3.3.1 **GUI vs. API**: Check this box if the humans had access to a graphical UI and the AIs only had API inputs
   *Checkbox item*

   3.3.2 **UI Equivalence Adjustment**: If no: does the eval attempt to adjust for the differences?
   *Select one of: "Yes", "Partial", "No" (Default), "Unknown/Unreported", or "N/A"*

3.4 **Instruction Equivalence**: Did the human baseliners and AIs have access to the same instructions/prompt/question for each item?
   *Select one of: "Yes", "Partial", "No" (Default), "Unknown/Unreported", or "N/A"*

   3.4.1 **Instruction Equivalence Adjustment**: If no: does the eval attempt to adjust for the differences?
   *Select one of: "Yes", "Partial", "No" (Default), "Unknown/Unreported", or "N/A"*

3.5 **Tool Access Equivalence**: Did the human baseliners and AIs have access to the same (technical) tools for each item? Respond yes if neither group had access to extermal tools; respond yes if the human had internet access and the AI did not (but was trained on the internet)
   *Select one of: "No" (Default), "Partial", "No", "Unknown/Unreported", or "N/A"*

   3.5.1 **Tool Access Equivalence Enforcement**: If human baseliners' tool access was limited: was there an oversight mechanism for ensuring that the human baseliners only used the tools permitted? E.g., enforcement of AI tool use ban
   *Select one of: "Yes", "Partial", "No" (Default), "Unknown/Unreported", or "N/A"*

3.6 **Explanations**: Did the eval/baseline collect explanations from the human baseliners, after the evaluation was conducted? I.e., explanations for why the human participants responded the way they did
   *Select one of: "Yes", "Partial", "No" (Default), "Unknown/Unreported", or "N/A"*

A.4   BASELINE ANALYSIS

4.1 **Statistical Significance**: Did the eval test for statistically significant differences between AI and human performance?
   *Select one of: "Yes", "Partial", "No" (Default), "Unknown/Unreported", or "N/A"*

   4.1.1 **Statistical Significance Test**: If yes: what statistical test was used?

4.2 **Uncertainty Estimate**: Did the paper present a measure of uncertainty for the AI and human baseline results? E.g., confidence intervals, variance, pooled/clustered standard errors, etc.?
   *Select one of: "Yes", "Partial", "No", "Unknown/Unreported", or "N/A"*

   4.2.1 **Estimate Type**: Is the reported baseline a point estimate, an interval estimate, or a distribution?
   *Select all that apply*
   • Point estimate (Default)

- Interval estimate
- Distribution estimate

4.3 **Evaluation Metric Equivalence**: Was the same evaluation metric measured/compared for both humans and AIs? Respond "no" if, e.g., the human baseline is majority vote but the AI baseline is not
*Select one of: "No" (Default), "Partial", "No", "Unknown/Unreported", or "N/A"*

4.4 **Evaluation Scoring Criteria Equivalence**: Was the same scoring rubric used for both AI and human results?
*Select one of: "No" (Default), "Partial", "No", "Unknown/Unreported", or "N/A"*

4.5 **Evaluation Scoring Method Equivalence**: Was the same scoring method used for both AI and human results? E.g., human grading, LLM as a judge
*Select one of: "No" (Default), "Partial", "No", "Unknown/Unreported", or "N/A"*

4.6 **Quality Control Robustness**: If quality controls were implemented: are analyses robust to different choices of exclusion criteria? E.g., do the authors state that the results don't change when including/excluding incomplete data?
*Select one of: "Yes", "Partial", "No", "Unknown/Unreported", or "N/A"*

## A.5   BASELINE DOCUMENTATION

5.1 **Additional Reporting**: Were the following reported?

5.1.1 **Reporting Sample Demographics**: Demographics for human baseliners, e.g., race, gender, etc. Respond yes only if within-sample demographics are reported; e.g., respond no if the paper only reports that 100% of the sample is based in the U.S.
*Select one of: "Yes", "Partial", "No" (Default), "Unknown/Unreported", or "N/A"*

5.1.2 **Reporting Baseline Instructions**: Instructions/guidelines given to human baseliners
*Select one of: "Yes", "Partial", "No" (Default), "Unknown/Unreported", or "N/A"*

5.1.3 **Reporting Time to Completion**: Time to completion for the eval items
*Select one of: "Yes", "Partial", "No" (Default), "Unknown/Unreported", or "N/A"*

5.1.4 **AI Tool Versions**: AI tools and versions (if baseliners had AI access)
*Select one of: "Yes", "Partial", "No" (Default), "Unknown/Unreported", or "N/A"*

5.1.5 **Completion Rate**: How many human baseliners were recruited but did not complete the tasks?
*Select one of: "Yes", "Partial", "No" (Default), "Unknown/Unreported", or "N/A"*

5.2 **Baseline Data Availability**: Is the (anonymized) human baseline data publicly available?
*Select one of: "Yes", "Partial", "No" (Default), "Unknown/Unreported", or "N/A"*

    5.2.1 **Individual Baseline Data Availability**: If yes: is data available at the individual baseliner level? I.e., can you tell from the dataset which baseliners were responsible for which questions?
*Select one of: "Yes", "Partial", "No" (Default), "Unknown/Unreported", or "N/A"*

    5.2.2 **Baseline Data Non-Availability Justification**: If no: is there a reasonable justification for non-disclosure of the baseline dataset? E.g., privacy concerns, safety/security concerns, company policy, etc.

5.3 **Experimental Materials Availability**: Are experimental materials used to implement the eval/baseline publicly available?
*Select one of: "Yes", "Partial", "No" (Default), "Unknown/Unreported", or "N/A"*

5.4 **Analysis Code Availability**: Is the code used to analyze the eval/baseline publicly available?
*Select one of: "Yes", "Partial", "No" (Default), "Unknown/Unreported", or "N/A"*

# B  APPENDIX: METHODOLOGY

We adopted a two-stage methodology as described in Section 3, adapted from the methodology of Zhao et al. (2025) and Reuel et al. (2024).

Section B.1 describes stage one, in which we conducted a meta-review of the measurement theory and AI evaluation literatures to qualitatively synthesize the checklist in Appendix A.

Section B.2 describes stage two, in which we systematically reviewed human baselines in foundation model evaluations.

## B.1  META-REVIEW

We begin with a scoping meta-review (a review of reviews) to qualitatively identify and synthesize literature relevant to human baselining. Meta-reviews are useful when there is little direct literature on the research question of interest (here, human baselines) but there is relevant literature from related fields (here, measurement theory) (Sarrami-Foroushani et al., 2015). As there is a wealth of literature in measurement theory, a meta-review that synthesizes the relevant evidence is appropriate to collect evidence in one place and to prevent researchers from being overwhelmed by the quantity of evidence (Hennessy et al., 2019).

Our literature search process adopted a purposive sampling approach. Although a systematic search process is normally ideal (Hennessy et al., 2019), purposive sampling is also acceptable for qualitative literature synthesis (e.g., Ames et al. 2019) and is justified here due to the broad scope of the relevant literature (Palinkas et al., 2015). Our sampling approach used theory-based inclusion criteria (Palinkas et al., 2015): we queried Google Scholar and Annual Reviews (2025a) in December 2024 for the keywords in Table 1, then filtered according to the criteria in Table 1. We also conducted backwards snowballing for the ML articles to identify further relevant literature. Finally, we added items to the sample based on our expertise, as many of the authors have experience in social science methodology and AI evaluation.

One limitation of this search strategy is that it introduces some sampling bias due to searching directly on the Annual Review website. We consider this limitation acceptable because by impact factor, Annual Reviews is a top-ranked publisher of literature reviews in the relevant social science disciplines (e.g., political science, psychology, sociology, statistics, economics) (Annual Reviews, 2025b). We thus expect our meta-review sample to be high-quality and relatively high-coverage.

Our search process yielded a total of 29 articles to be included in our meta-review (listed in Table 2). To synthesize our checklist, KW scanned these 29 articles and compiled a list of relevant methodological practices/considerations in a Google Sheet, categorizing each into the categories of baseline(r) design, recruitment, implementation, analysis, and documentation. The authors then collectively discussed the checklist and validated the checklist using expert feedback from six external experts before refining and finalizing the checklist. Finally, the checklist was also iteratively refined during the coding process.

Table 1: Inclusion criteria for meta-review articles.

| Type | Inclusion Criteria |
|---|---|
| Document type | Literature review
Position paper
Synthesis article
Book or book chapter (including reference texts) |
| Subject area | Measurement theory (including applications in statistics, economics, political science, psychology, education, sociology, or medicine)
AI evaluation |
| Keywords (non-exhaustive) | "measurement theory"
"measurement model*"
"validity"
"reliability"
"replicability"
"survey design"
"survey method*"
"questionnaire design"
"experimental design"
"causal inference" |

Table 2: A complete list of the 29 articles included in our meta-review.

| Subject area | Articles |
|---|---|
| Measurement theory ($n = 17$) | Bandalos (2018); Berinsky (2017); Cai et al. (2016); Chang et al. (2021); Couper (2017); Findley et al. (2021); Groves et al. (2011); Imbens & Rubin (2015); Jackson & Cox (2013); Kertzer & Renshon (2022); List et al. (2011); Nosek et al. (2022); Rosellini & Brown (2021); Stantcheva (2023); Strauss & Smith (2009); Zhang et al. (2023); Zickar (2020) |
| Machine Learning ($n = 12$) | Agarwal et al. (2022); Bowman & Dahl (2021); Cowley et al. (2022); Dow et al. (2024); Eckman et al. (2025); Ibrahim et al. (2024); Liao et al. (2021); Reuel et al. (2024); Subramonian et al. (2023); Wang et al. (2023); Xiao et al. (2023); Zhou et al. (2022) |

## B.2 SYSTEMATIC LITERATURE REVIEW

We conducted a systematic literature review of human baselines in AI evaluations (Page et al., 2021) to identify gaps in baselining methodology. Our review method is the similar to that of Zhao et al. (2025).

We conducted a systematic search for relevant literature. To begin, we queried Google Scholar in December 2024 for articles containing the keywords in Table 3. Our search terms were intentionally broad, as authors use a variety of different language to describe human baselines. Articles were included in the initial sample if they contained in the full text both a human baseline keyword and an AI evaluation keyword.

Table 3: Search terms for systematic literature review of human baselines

| Type | Keywords |
|---|---|
| Human Baseline Keywords | "human baseline*"
"expert baseline*"
"human performance baseline*" |
| AI Evaluation Keywords | "LLM evaluation*"
"AI evaluation*"
"NLP evaluation*"
"ML evaluation*"
"model evaluation*"
"LLM benchmark*"
"AI benchmark*"
"NLP benchmark*"
"ML benchmark*"
"evaluating LLM*"
"evaluation of LLM*"
"benchmark LLM"
"benchmarking LLMs"
"evaluation of AI models" |

Google Scholar was chosen as the database of choice due to its comprehensive coverage (Gusenbauer, 2019) and its indexing of the gray literature. We included articles in the gray literature (e.g., preprints) because researchers often post preprints on arXiv prior to formal publication and because a substantial portion of ML literature is published on arXiv, including publications from many industry organizations (Shah Jahan et al., 2021). For instance, arXiv was the source of an overwhelming majority of articles in one recent systematic literature review on bidirectional language models (Shah Jahan et al., 2021).

There is debate in the methodological literature about the use of Google Scholar as a primary database in a systematic literature review. Concerns have been raised about limitations to advanced search capabilities and to the Google Scholar interface (Halevi et al., 2017), lack of precision (Boeker et al., 2013), and lack of coverage (Haddaway et al., 2015). We addressed these limitations as follows:

- To address limitations to advanced search capabilities, we did not use advanced search capabilities beyond the boolean AND and OR operators in search strings, as well as a simple date filter.
- To address interface limitations, we created workarounds by using multiple queries (to avoid the 256 character limit in search strings) and using a bookmarklet to capture reference information. In any case, we generally find that the search capabilities and interface of Google Scholar are an improvement over the search function in arXiv, making queries to Google Scholar preferable to direct queries in arXiv.
- To address limitations in precision, we adopted more stringent inclusion/exclusion criteria to filter our sample (discussed below). Furthermore, our search is necessarily imprecise due to a lack of standardization of terms in describing human baselines in the literature (e.g., we found that some literature described baselines as "human evaluation", which is normally used to describe human annotations of evaluation data).

- To address limitations in coverage, we supplemented our Google Scholar search with other sources. SD queried Elicit for articles containing human baselines, and MB identified evaluation datasets with human baselines used in industry evaluations by scanning the model/system cards of cards of OpenAI o1 (OpenAI et al., 2024), Anthropic's Claude 3.5 Sonnet (Anthropic, 2024a), Meta's Llama 3 (Grattafiori et al., 2024), and Google DeepMind's Gemini 1.5 (Gemini Team Google et al., 2024).[3] Furthermore, the most recent research has found that Google Scholar has significantly expanded its coverage (Gusenbauer, 2019), and another study found that Google Scholar indexed 96% of articles in systematic literature reviews in computer science that were conducted using other databases (Yasin et al., 2020).

Our search process yielded a sample of $n = 397$ articles (378 from Google Scholar, 13 from Elicit, and 6 from industry model/system cards), which were stored in a Google Sheet. KW then scanned the title, abstract, and main text of each article to filter the sample; the inclusion/exclusion criteria used in filtering along with rationales for each criterion are discussed in Tables 4 and 5. As Google Scholar does not always index the most authoritative version of articles, KW also cross-referenced DBLP for all articles on preprint servers (including arXiv) to identify the latest version or published version of each preprint. During the coding process, all coders were also made aware of the exclusion criteria in case any invalid articles were inadvertently included for coding. The final number of articles included for analysis was $n = 109$, and these are identified in Table 6.

Table 4: Inclusion criteria for systematic review of human baselines.

| Inclusion Criteria | Rationale |
|---|---|
| Article contains an evaluation of a foundation model | • We limited our scope to foundation models in part to make the review practically manageable
• No comprehensive guidance exists for human baselines that is specific to the context of foundation models and that accounts for the most recent foundation model literature
• Foundation models raise different and somewhat unique considerations for human baselines, and we aimed to narrow in on these specific considerations
• Examples of qualifying articles: articles that fine-tuned or used pre-trained large (language or multi-modal) models |
| Article contains a human baseline (defined in Section 1) | • See exclusion criteria for examples of non-qualifying articles |
| Article is published in a peer-reviewed venue or is available in the gray literature (e.g., on a preprint server such as arXiv) | • See text for a discussion of arXiv |

We then coded each of the included articles per our checklist, with results stored in a Google Sheet. Following the coding strategy in Zhao et al. (2025), a subset of authors each coded the same four articles, discussed results to ensure coding consistency, and refined the checklist items. The remaining articles were then split up for coding between authors. Questions that arose during the final coding process were adjudicated via discussion, and KW cleaned and standardized the final dataset.

Note that many articles conducted human baselines for multiple different datasets. During the coding process, each dataset for which a baseline was conducted was coded separately. During the process of cleaning and analyzing coded data, only baselines that contained key differences in baseline

---

[3]The date filter in Table 5 was not applied for these articles so that we could capture evaluations that are widely used in practice. Only one article that would have otherwise been excluded was ultimately included in our sample of baselines (Dua et al., 2019).

Table 5: Exclusion criteria for systematic review of human baselines.

| Exclusion Criteria | Rationale |
| --- | --- |
| AI model being evaluated is not a foundation model | • See inclusion criteria for discussion of rationale
• Examples of non-qualifying articles: articles that trained non-general purpose models for specific purposes |
| Article did not contain a human baseline | • Enforcement of analogous inclusion criteria |
| Human baseline in the article is not original (e.g., uses real-world data or a human baseline from a pre-existing dataset) | • Articles using real-world data are excluded as it is difficult to make direct comparisons between human and AI performance in such cases, given that the human data was not generated in a controlled laboratory setting
• Articles using pre-existing human baseline data are excluded as researchers may fail to adhere to the experimental design of the previous baseline, making comparisons difficult |
| Article duplicates an item already included in the review | • Prevention of duplicate items
• Examples of non-qualifying items: preprint or workshop version of subsequently published work |
| Article was published before 2020 | • Most foundation model evaluation literature was published after 2020 (inclusive) |
| Article collected data from human annotators but not as a baseline | • Use of human data in non-baseline contexts gives rise to different methodological considerations
• Examples of non-qualifying articles: articles using human evaluation (i.e., using human annotators to score or analyze evaluation data), articles collecting human data as ground truth (e.g., using annotations to determine the desired responses to evaluation items) |
| Article evaluates LLM-as-a-judge, i.e., compares LLM vs. human evaluation of AI models | • LLM-as-a-judge may give rise to highly idiosyncratic methodological considerations |
| Article is incomplete work or work-in-progress | • Quality control
• Examples of non-qualifying articles: articles submitted to venues but not released as preprints (e.g., paper available on OpenReview but not on arXiv; we assume that authors of these articles do not intend to make their papers public), articles submitted to non-archival workshops intended to refine work |
| Article is a thesis or class work | • Quality control |

instrument, construct, sample, or process were retained as distinct baselines; these were formalized as different coding for Q1.5–1.8, Q2.1–2.3, Q2.6–2.7, or Q3.2–3.5. We find four articles which contained more than one distinct baselines by this criteria (Lu et al., 2024; Meister et al., 2024; Castro et al., 2022; Verma et al., 2024), bring the total number of human baselines up to 113.

Table 6: A complete list of the 109 articles included in our systematic review of human baselines. Note that we analyze 113 individual baselines from these articles, as a single article may contain multiple baselines (see explanation in text).

|  | **Articles** |
|---|---|
| Included ($n = 109$) | Abdibayev et al. (2021); Akhtar et al. (2024); Albrecht et al. (2022); Alex et al. (2021); Asami & Sugawara (2023); Asiedu et al. (2025); Awal et al. (2025); Bai et al. (2024); Blinov et al. (2022); Bu et al. (2024); Castro et al. (2022); Chang et al. (2024a; 2023); Chen et al. (2024); Chhun et al. (2024); Chiu et al. (2024); Chiyah-Garcia et al. (2024); Costarelli et al. (2024); Dagli et al. (2024); Dua et al. (2019); Duan et al. (2022); Fenogenova et al. (2024); Fyffe et al. (2024); Gong et al. (2024); Gu et al. (2024); Guo et al. (2024); Gupta et al. (2024); Haan et al. (2024); Hackenburg et al. (2023); Hamotskyi et al. (2024); Heiding et al. (2024); Hendrycks et al. (2021b); Hijazi et al. (2024); Hildebrandt et al. (2024); Hou et al. (2024); Huang et al. (2024); Ivanov (2024); Jain et al. (2023); Ji et al. (2022); Jimenez et al. (2022); Jing et al. (2023); Kodali et al. (2024); Kruk et al. (2024); Lacombe et al. (2023); Laine et al. (2024); Laurent et al. (2024); LeGris et al. (2024); Lei et al. (2024a); Li et al. (2024a;b; 2021; 2025); Lin et al. (2022); Liu et al. (2024a; 2023; 2024b); Lu et al. (2024; 2023); Mangalam et al. (2023); Meister et al. (2024); Mialon et al. (2023); Miller et al. (2020); Mirza et al. (2024); Mizrahi et al. (2020); Montalan et al. (2024); Moskvichev et al. (2023); Mukhopadhyay et al. (2024); Norlund et al. (2021); Obeidat et al. (2024); Phuong et al. (2024); Reese & Smirnova (2024); Rein et al. (2024); Roberts et al. (2024); Ruis et al. (2023); Sakai et al. (2024); Santurkar et al. (2020); Sanyal et al. (2024); Shavrina et al. (2020); Si et al. (2024); Someya & Oseki (2023); Sourati et al. (2024); Sprague et al. (2023); Srivastava et al. (2023); Suvarna et al. (2024); Tahsin Mayeesha et al. (2021); Taktasheva et al. (2022); Tam et al. (2024); Tanzer et al. (2023); Thrush et al. (2024); Valmeekam et al. (2023); Verma et al. (2024); Wadhawan et al. (2024); Webson et al. (2023); Weissweiler et al. (2024); Wijk et al. (2024); Wu et al. (2024; 2023); Xiang et al. (2023); Yin et al. (2024); Yue et al. (2024); Zamecnik et al. (2024); Zerroug et al. (2022); Zhang et al. (2024a;b;c); Zhou & Hong (2024); Zhou et al. (2024); Zhu et al. (2023); Zhuo et al. (2024) |

## C  APPENDIX: DATA AVAILABILITY

An up-to-date version of our checklist as well as individual annotations from our systematic review of human baselines are available at `https://github.com/kevinlwei/human-baselines`

## D  APPENDIX: ALTERNATIVE VIEWS

We discuss four alternative views to our position below.

**Alternative View 1: Human baselines will soon become unnecessary or insufficient for many evaluations as AI models surpass expert human performance Goldstein & Sastry (2024).** Human baselines—in addition to other AI baselines—may be useful even if AI models surpass expert human performance. For instance, they can determine the *magnitude* of human vs. AI performance differences, which is important for modeling economic impacts and making for business or policy decisions Eloundou et al. (2023). They can also help researchers understand how cognition and behavioral tendencies differ between humans and AI systems. Additionally, many authors report random baselines, which can assist interpretation of evaluation results; at the very least, a human baseline could serve as a floor for expected performance from foundation models.

**Alternative View 2: Existing human baselines or real-world data may be enough to measure progress, even if only approximately.** Some existing baselines may meaningfully measure performance, but many are insufficiently rigorous to draw conclusions about the pace of AI progress Tedeschi et al. (2023); Cowley et al. (2022). Moreover, stakeholders may demand additional rigor for evaluations used in, e.g., risk assessments or safety cases Goemans et al. (2024). Secondary data like standardized tests can be useful points of comparison by providing score distributions from large samples, but it may not always exist for desired use cases. Secondary data is also less well-validated for evaluating models: data contamination concerns are common Yao et al. (2024) and models can perform strangely on assessments designed for humans (e.g., Lei et al. 2024b).

**Alternative View 3: Implementing all the practices in this checklist is too expensive to be realistic.** We agree that collecting high-quality baseline data can be prohibitively costly. Our aim is not to provide hard requirements for authors to follow but rather an instructional framework to help researchers understand the impact of methodological design choices, allowing researchers to judge whether the rigor provided by particular design choices is justified by the marginal cost and by the evaluation's intended use case. Even where researchers decline to opt for more rigorous methods, reporting study details can nevertheless improve transparency and enable external assessments of published baselines. In general, we believe that the value of more rigorous and transparent human baselines is sufficiently high that funders and the ML community should establish more stringent norms for scientific rigor in AI evaluations.

We also believe that researchers can make many low-cost improvements to human baselining methods—for instance, carefully selecting baseline test sets, reporting uncertainty or statistical tests, and avoiding baselines performed by authors previously exposed to baseline items. Cost considerations are also not unique to ML and have been widely acknowledged in, e.g., survey methodology Leeuw (2005); methods in other fields such as phased clinical trials were developed in part to account for budgetary considerations.

**Alternative View 4: This framework and checklist may not be appropriate in all cases due to differing needs in human baselines.** Our framework and checklist are not meant as one-size-fits-all recommendations; different evaluations and contexts require different baselining methods. Our intention is to provide an informational guide for designing and assessing baselines (see Alternative View 3 and Section 4). Furthermore, we believe that some standardization—common in many other fields Winters et al. (2009)—is nevertheless useful for transparency, replicability, and interpretability of results (see Kapoor et al. 2024).

