# OpenReview forum: "Model Evaluations Need Rigorous and Transparent Human Baselines"
_ICLR.cc/2025/Workshop/BuildingTrust — BuildingTrust_

### Official Review · Reviewer_9kLM · 2025-02-26
**A nice position paper**

**Rating:** 7
**Confidence:** 4

**Review:**

The paper is well written and insightful. It analyzes and discusses many aspects in successful human baselines for the evaluation of foundation models. The literature research is also extensive.

Small comments to the authors:
- Lines 326-328 seem contradictory to what you say in paragraph 305-312, maybe it is worth it to make your point more clear here.
- Lines 334-336 seem like a left-in comment and not part of the text?
- Line 338 "turnbe" typo
- Line 397 "andresources" typo
- Line 402 I think a comma is missing

---

### Official Review · Reviewer_nBXj · 2025-02-28

**Rating:** 7
**Confidence:** 3

**Review:**

Summary:

The paper is a position paper, which argues that human performance baselines in AI evaluations must be more rigorously designed and transparently reported to enable meaningful comparisons. The authors highlight that many AI models claim superhuman performance, but the methods used to establish human baselines often lack rigor, such as small or biased sample sizes, inadequate controls for confounding variables, and inconsistent measurement frameworks. To address these shortcomings, the paper proposes a framework based on measurement theory. It assesses the validity and reliability of human baseline methods and was used to review 113 existing human baseline studies in AI evaluations. The authors provide a checklist for researchers to improve the design, implementation, and reporting of human baselines, aiming to enhance the trustworthiness of AI performance assessments.

Strengths:

- The paper discusses a critical issue in AI evaluation, emphasizing the need for rigorous and transparent human baselines. This is particularly important for trustworthyness given the increasing claims of AI systems outperforming humans in various tasks.

- The proposed framework, grounded in measurement theory, provides a structured approach to evaluating and improving human baseline methodologies. The checklist also offers practical guidance for researchers.

Weaknesses:

- While the framework is rigorous, implementing it fully seems to be resource-intensive. A discussion on cost-effective strategies for smaller research teams or practitioners could improve its accessibility.

Overall Evaluation:

This is a well-researched and timely position paper, and it fits the workshop, I recommend accept.

---

### Official Review · Reviewer_XY4u · 2025-03-02

**Rating:** 7
**Confidence:** 2

**Review:**

This is a well-crafted position paper that thoughtfully tackles an important issue in the field. It nicely articulates the problem of inconsistent and poorly defined human baselines in foundation model evaluations, shedding light on a critical challenge that deserves more attention.
It’s great to see the authors’ detailed analysis of 113 papers which identifies recurring problems.
Their systematic review exposes key shortcomings, such as inconsistencies in test sets and weak sampling methods, providing a clear picture of the flaws in current practices.
It’s nice to see actionable tools are proposed like the checklist, as it offers researchers a structured way to improve the quality and transparency of human baselines in future AI studies.

---

### Decision · Program_Chairs · 2025-03-04

**Decision:**

Accept

**Comment:**

This well-crafted position paper highlights the inconsistencies in human performance baselines for AI evaluations, emphasizing the need for rigor and transparency to ensure meaningful comparisons. It introduces a measurement theory-based framework and checklist to improve baseline design, though implementation may be resource-intensive for smaller research teams.